# Hydroxyurea Pharmacokinetic Evaluation in Patients with Sickle Cell Disease

**DOI:** 10.3390/ph17101386

**Published:** 2024-10-17

**Authors:** Daniela Di Grazia, Cristina Mirabella, Francesco Chiara, Maura Caudana, Francesco Maximillian Anthony Shelton Agar, Marina Zanatta, Sarah Allegra, Jenni Bertello, Vincenzo Voi, Giovanni Battista Ferrero, Giuliana Abbadessa, Silvia De Francia

**Affiliations:** 1Laboratory of Clinical Pharmacology “Franco Ghezzo”, Department of Clinical and Biological Sciences, University of Turin, 10043 Orbassano, Italy; daniela.digrazia@unito.it (D.D.G.); cristina.mirabella@edu.unito.it (C.M.); francesco.chiara@unito.it (F.C.); mauracaudana98@gmail.com (M.C.); francescomaximilliananthony.sheltonagar@unito.it (F.M.A.S.A.); sarah.allegra@unito.it (S.A.); giuliana.abbadessa@unito.it (G.A.); 2Department of Economics and Statistics “Cognetti de Martiis”, University of Turin, 10124 Turin, Italy; m.zanatta1@campus.unimib.it; 3Microcythemia and Rare Haematological Diseases Center, Department of Clinical and Biological Sciences, University of Turin, 10043 Orbassano, Italy; jennibertello@gmail.com (J.B.); vincenzo.voi@unito.it (V.V.); giovannibattista.ferrero@unito.it (G.B.F.)

**Keywords:** therapeutic drug monitoring, variability, maximum tolerated dose, area under the curve, high-pressure liquid chromatography, sex

## Abstract

**Background:** Hydroxyurea (HU), also known as hydroxycarbamide, is an oral ribonucleotide reductase inhibitor. In 1999, the United States Food and Drug Administration (FDA) approved HU for the treatment of sickle cell disease (SCD). Since then, it has become the cornerstone in the management of SCD patients, helping to reduce vaso-occlusive crises, acute chest syndrome, the need for blood transfusions, hospitalizations and mortality. There is considerable variability among individuals in HU pharmacokinetic (Pk) parameters that can influence treatment efficacy and toxicity. The objective of this work is part of a clinical study aimed at investigating HU Pk and determining the optimal sampling time to estimate the Area Under the Curve (AUC) in SCD patients. **Methods:** HU plasma concentration in 80 patients at various time points (2, 4, 6, 24 h) following a 48-h drug washout was quantified using High-Pressure Liquid Chromatography (HPLC) coupled with an ultraviolet (UV) detection method previously described in the literature and adapted to new conditions with partial modifications. **Results:** The mean HU administered dose was 19.5 ± 5.1 mg/kg (range: 7.7–37.5 mg/kg). The median AUC quantified in plasma patients was 101.3 mg/L/h (Interquartile Range (IQR): 72.5–355.9) and it was not influenced by the weight-based dose. However, there was a strong positive correlation between AUC and Body Mass Index (BMI) as well as dose per Body Surface Area (BSA). Along with a three-point approach for AUC determination present in the literature, we show results obtained from a four-point sampling strategy, which is more useful and effective for better optimizing dose escalation to the maximum tolerated dose (MTD). Moreover, we observed that most patients achieved the maximum HU plasma concentration two hours after drug administration, regardless of age differences. **Conclusions:** HU treatment, which represents a milestone in the treatment of SCD due to its ability to reduce disease complications and improve patients’ quality of life, requires careful monitoring to optimize the individual dose for saving potential side effects and/or adverse events.

## 1. Introduction

SCD is a hemoglobin (Hb)-related blood disorder that is typically inherited. It is mainly characterized by the formation of abnormal Hb long chains, ending in sickle-shaped red blood cells, multi-organ damage and increased mortality. In SCD, sickling and hemolysis of red blood cells lead to vaso-occlusion associated with ischemia. SCD is diagnosed through neonatal screening, if available, or when patients report inexplicable severe atraumatic pain or normocytic anemia [1]. Around the word every year, 300,000 infants are born with SCD, and they mainly live in southern countries, such as sub-Saharan Africa, India, the Mediterranean area and the Middle East. In 2007, the World Health Organization (WHO) estimated that 2.1% of the global population were carriers of SCD. Between 2000 and 2021, the total number of births of individuals affected by SCD increased by 13.7%, primarily due to demographic growth in the Caribbean and sub-Saharan Africa [2]. SCD consistently ranks among the top 20 causes of mortality for children under 5 years old, those aged 5–14 years, and individuals aged 15–49 years in over half of the world’s regions [3]. In Europe, the incidence of SCD has been rising due to migration and increased life expectancy. Despite this, it is still classified as a rare disease in Europe, affecting fewer than 5 people per 10,000. In Italy, the prevalence was estimated to be 13 per 100,000 people in 2018 [4]. This disorder is most commonly caused by a missense homozygous mutation at the sixth amino acid of the β-globin chain, leading to the expression of valine instead of glutamic acid. The mutated hemoglobin, known as HbS, is prone to polymerization when deoxygenated under stress conditions. This results in the sickle-like deformation of erythrocytes, which are less elastic, have severely disarranged rheological properties, and are accompanied by a chronically increased inflammatory state with elevated proinflammatory cytokines such as C-reactive protein, TNF, IL-1, and IL-8 [5,6]. Hematopoietic stem cell transplantation is the only curative option for SCD, but its use is limited by the unavailability of a matching donor, high costs, and the risk of graft-versus-host disease [7]. The treatment options for this condition are still very limited, including transfusion, exchange transfusion and HU. The first two improve microvascular flow and reduce endothelial injury and inflammation but increase the risks of iron overload, alloimmunization and transfusion reactions. HU remains a readily available and economical solution [1]. Initially synthesized in 1869, HU did not find clinical use until the 1960s, when its effectiveness in treating myeloproliferative disorders was established. In the 1980s, HU was considered as a potential treatment for SCD and it was approved by FDA for SCD in 1999 [8]. HU is highly hydrophilic, with a volume of distribution similar to water. Approximately 37% of ingested HU is excreted unmetabolized by the kidneys, with this percentage rising to about 50% in children. Some of the drug is metabolized in the liver into urea and carbon dioxide by cytochrome P450 monooxygenases, while a portion is converted into nitric oxide (NO).

Although it is widely used, the mechanism of action of HU is not fully understood, and significant variability in the Pk profile, pharmacological response, and tolerance has been observed among patients [9,10,11,12,13]. Differences in the Pk of HU have been noted in SCD patients, particularly during the absorption and elimination phases. Specifically, the common occurrence of kidney disease in SCD patients can alter HU renal clearance, either increasing or decreasing it [14,15]. Therefore, it is essential to address Pk variability through pharmacokinetic/pharmacodynamic (Pk/Pd) studies and therapeutic drug monitoring (TDM) applications. TDM is the only quantitatively useful tool to justify the appearance of side effects in patients. Common side effects for SCD patients of HU include eczema, xeroderma, macrocytosis, neutropenia and subsequent infections. Myelosuppressive effects can be mitigated by adjusting the dosage and are considered markers for reaching the MTD. And, even in this case, TDM can prove to be of great use. Overall, early and sustained HU treatment has shown a positive impact on both morbidity and mortality [16]. As regards the management of HU administration, SCD infants are generally advised to start treatment at 9 months of age. Despite its short half-life, HU is usually administered once daily. It is almost fully bioavailable orally and available in both liquid and tablet forms, considered equivalent with minimal differences in absorption rates. European centers often prescribe a fixed dose of 20 mg/kg/day for all patients, regardless of potential for higher fetal hemoglobin (HbF) induction with increased dosages. This approach is partly due to challenges in monitoring dose escalation side effects and concerns about drug safety among patients and healthcare providers [16]. 

In the dose escalation method, treatment begins with 20 mg/kg/day, with the aim of reaching the MTD to maximize benefits. This approach requires regular follow-up visits and lab tests, which can affect patient compliance, though follow-up intervals may become less frequent over time. The MTD is determined by increasing the dose while ensuring that absolute neutrophil count (ANC) remains above 3.0 × 10^9^/L, Hb above 5.0 g/dL, reticulocytes above 100 × 10^9^/L, and platelets above 100 × 10^9^/L. Once these thresholds are met or exceeded, dose escalation is halted [9]. This method is considered safe with appropriate monitoring, as side effects can be reversed by dose reduction. Managing SCD at MTD has been shown to offer the best benefit–risk ratio, although it requires a prolonged period of 6 to 12 months. If no alternatives are available, optimizing the benefits and time to reach MTD continue to be important goals [9,17]. Precision medicine protocols have been developed to enhance individual response targets [18]. Dossantos Neres et al. highlighted how different studies use various sampling and estimation methods to accurately determine Pk parameters while minimizing patient distress, which is particularly challenging for children involved in these studies [17]. Optimizing sampling times, the number of samples, and the volume of blood drawn are crucial for making Pk-guided dosing more practical and widespread. But, there is not much scientific literature available for this purpose.

In light of this, the main objective of the present study was to develop, based on the limited existing literature, an HPLC-UV method for quantifying HU plasma concentration, in order to achieve the following:To investigate the main resulting Pk parameters after a single administered dose of HU in SCD patients;To suggest a useful tool for clinicians to determine the optimal sampling time strategy to estimate the AUC.

Finally, another objective of the study was to evaluate possible differences in HU AUC plasma levels between male and female patient populations. Important sex and gender differences are observed in the outcome (frequency, symptoms and severity) of common diseases. Also, response to treatments and adverse drug events may be affected by these differences, but they are still underestimated. A sex- and gender-based approach applied to clinical practice can significantly contribute to health promotion, increasing therapeutic appropriateness and improving benefits for patients and for the National Health System. Also, in treating patients with SCD, this approach can be usefully applicable [19]. We only considered sex differences for our patients’ population due to the lack of data collected related to gender, a purpose to be implemented in terms of data collection in the future.

## 2. Results

### 2.1. Methodology from Legrand et al. and Calibration Curves

The method adapted from the literature is that of Legrand et al. [20]. To validate the adapted methodology to the available instrumentation and perform a reliable quantification of HU plasma levels, different sets of calibration curves wereprepared. For the calibration curve, we prepared a series of standards at known and increasing concentrations of HU: 0.625 μg/mL-1.25 μg/mL-2.5 μg/mL-5 μg/mL-10 μg/mL-20 μg/mL-40 μg/mL. An example of the HU calibration curve is shown in Figure 1. A new calibration curve was produced for each group of patients analyzed, for a total of 10 curves. The mean of the total regression coefficients (R^2^) obtained from the R^2^ of each calibration curve was 0.9957. A representative HU calibration curve is reported in Figure 1.

Figure 2 shows an example of HU peaks obtained from the HPLC-UV analysis. The retention times of the xanthydrol (Xan) derivatives HU (Xan–HU) (Figure 2c,d) and Methyl-urea (Xan-Me-U) (Figure 2b–d) were approximately 6.1 and 10.8 min, respectively. The Xan derivative of the endogenous urea (Xan-urea) in the plasma showed a peak at 7.5 min (Figure 2a–d). Two other peaks corresponding to the unconjugated Xan were observed at 11.7 and 15.4 min (Figure 2a–d).

### 2.2. Population Enrolled

Eighty different SCD patients treated with HU were enrolled between July 2023 and March 2024 (none were recruited more than once). The mean HU administered dose was 19.5 ± 5.1 mg/kg (range: 7.7–37.5 mg/kg) and it was not influenced by the weight-based dose. In total, 52.2% of patients were female (N = 42) and 47.8% (N = 38) were male (Table 1). In Table 1, the patients’ data are expressed by median values with IQR.

In total, 72.5% of patients were of African origin, 21.3% were Caucasian and 6.3% were classified in the Other group (mixed heritage) (Table 2).

Overall, 10% of patients (N = 8) were naïve to HU at the beginning of the study, while 90% (N = 72) were on treatment with HU for at least 60 days. Patients in the steady-state treatment had been taking HU for 7.3 ± 5.8 years on average (min = 0.23; max = 25.7) and 90.3% (N = 65) of them were on treatment with the MTD.

No patient started treatment before they were 2 years old, while the average age at the beginning of treatment was 10.9 ± 10.4 years (min = 2.3; max = 49.3).

### 2.3. HU Pk and Correlation with Age, BMI, BSA and Ethnicity

We tested 80 patients for a total number of 400 plasma determinations, considering the five different sampling points (0 prior to drug administration; 2, 4, 6, 24 h after drug intake). Each patient’s five-point curve was repeated twice to ensure a reliable accuracy in the HPLC-UV determination, for a total of 800 HU quantifications performed. We tested the HU Pk parameters of all patients between July 2023 and April 2024. In Table 3, the AUC, half-life (T_1/2_), elimination constant (K_e_), maximum concentration (C_max_) and time to reach maximum concentration (T_max_) expressed by median values with IQR are shown. Figure 3 represents the HU AUCs expressed by the median values obtained for each sampling point.

In Table 4, the HU AUCs achieved from all patients’ plasma and sex-disaggregated HU AUCs expressed by median values with IQR are shown.

We performed a *T*-test between age and the AUC; the age was proportional (<0.001) to the AUC even if corrected for dosage (indicated as AUCmk). The AUCmk increased by 1 mg/L/h/kg every 10 years, with a coefficient of 0.1.

Another important evaluation was carried out related to BMI as a predictor of AUC. There was a coefficient of 5.3 with a T-test of 0.019, as shown in Table 5; furthermore, BMI was significant regardless of age (Table 6). In Table 5 and Table 6, the values are expressed for the *T*-test coefficient by standard error (SE) and confidence interval (CI).

Considering the results, we decided to investigate how dose/BSA (mg/m^2^) correlates with AUC. In 80 patients enrolled, the average dose/BSA equaled 604.5 ± 135.7 mg/m^2^ (min = 299.3; max = 919.7) (Table 7, mean value of dose/BSA expressed by SD, min and max).

Table 8 shows that dose/BSA is an important factor for the AUC, with a coefficient of 0.2 and a *T*-test of 0.001 (value expressed for *T*-test coefficient by SE and CI).

In the case of patients on steady-state treatment, a lower AUC with a significant negative coefficient of −61.7 (Table 9) was determined for the African ethnicity. The coefficient was −5.3 even after adjusting for dosage (Table 10). In Table 9 and Table 10, values are expressed for the *T*-test coefficient by SE and CI.

### 2.4. Comparison between Methods

Comparing AUCs achieved from the application of the original methodology published in the study by Dong et al. [18] and those achieved by the use of the adapted methodology, no statistically significant difference was obtained, as indicated by a *T*-test value of 0.7334 (Table 11). In Table 11, the AUC mean values for the *T*-test are expressed by range of min and max, SD and CI.

### 2.5. Optimization Sampling Time to Estimate Reliable AUC

On the basis of all the HU levels quantified for each sampling point (0 prior to drug administration; 2, 4, 6, 24 h after drug intake), the clinicians proposed a feasible application for the determination of the optimal sampling times to estimate the AUC as accurately as possible, minimizing the number of blood withdrawals. A three-point formula (T2, T4 and T6) showed the strongest correlation with AUC (*T*-test < 0.0001 and a R^2^ = 0.98). Therefore, the first consideration was to estimate the AUC using these three points. AUC estimation using 2 points (T2 and T4) retained a *T*-test < 0.0001 but showed a lower R^2^, equaling 0.92. Ultimately, clinicians of the Hemoglobinopathies University Center of San Luigi Gonzaga Hospital proposed a reliable application of HU plasma determinations to estimate the AUC only using T2, which presented a *T*-test score < 0.0001 but showed the lowest r^2^ (0.86).

## 3. Discussion

In this study, we aimed to develop an HPLC-UV method, based on the existing literature [20], for quantifying HU plasma concentrations. The subject of the present study has been studied many times before, as indicated by the available scientific literature [9,10,11,12,13,15,16,17,18,20]. But, still no clear clinical tool is defined in the literature to adjust HU therapy in SCD patients by Pk analyses. Our goal was, then, to deeply investigate the Pk parameters, including C_max_, AUC, t_1/2_ and T_max_, of a single dose of HU administered in SCD patients, leading to a useful tool for clinicians to determine the optimal sampling time strategy to estimate the AUC. Despite minor modifications to the method, our Pk results were comparable to those obtained by Dong et al. [18], with no statistically significant difference (*T*-test = 0.733). We obtained an average AUC of 118.8 ± 66.6 mg/L/h, ranging from 42.4 mg/L/h to 355.9 mg/L/h, compared to the average AUC of 115.7 ± 34.0 mg/L/h ranging from 61.5 mg/L/h to 317.8 mg/L/h reported in their study. This correspondence is clear evidence of the robustness, statistical significance and validity of the developed technique. To adapt the methodology from the literature for reliable HU plasma level quantification, we prepared different sets of calibration curves. To obtain the calibration curve, we set a series of HU standards at known and progressively increasing concentrations from 0.625 to 40 μg/mL. The mean of the total R^2^ was 0.9957, confirming a good linear regression. In this study, 80 SCD patients treated with HU were enrolled between July 2023 and March 2024 (no patient was recruited more than once). Of these, 52.2% were female (N = 42) and 47.8% were male (N = 38), confirming a major disease incidence in the female population as indicated in the GREATALYS study (Generating Real world Evidence Across Italy In SCD, study code CSEG101AIT01). In the considered patient population, women treated with HU were younger than men (median age in years 11.6 vs. 17.4); women started HU treatment earlier than men (age in years 2.3 vs. 4.1); and women interrupted HU treatment earlier than men (age in years 55.1 vs. 73). This may be related to the characteristics of the patients’ followed at the Hemoglobinopathies University Center of San Luigi Gonzaga Hospital in Orbassano (Turin, Italy), but it would be interesting to further the study in terms of a sex approach, considering the potential influence of menopause in the progress of HU therapy. The majority of patients (72.5%) were of African origin, 21.3% were Caucasian and 6.3% were classified in the “Other” group (mixed heritage), as confirmed from WHO data. No patient started treatment before the age of 2 years, with the average age at treatment initiation being 10.9 ± 10.4 years (min = 2.3; max = 49.3). The study included both treatment-naïve and previously treated patients, who received HU after a 48 h washout. Plasma HU concentrations were determined at five different sampling points (0 h before drug administration, and 2, 4, 6, and 24 h after drug intake). Sex-disaggregated AUC data showed higher median levels achieved for males (117.9 mg/L/h; IQR 73.6–355.9) than those achieved in females (97.4 mg/L/h; IQR 71.9–336.8).

This observation could be viewed as a major indicator of the effectiveness of therapy in the male population, but further sex-disaggregated analyses are needed in terms of clinical parameters as well as increasing the number of patients enrolled. Further analyses will be needed to define the AUC values related to the efficacy and toxicity of HU treatment in patients, in order to better support clinicians in patient management.

By analyzing a HU steady-state population considered by ethnic origin, it is shown that patients of African ethnicity had a significantly lower AUC, with a coefficient of −5.3, even after dosage correction. Ethnic origin remained the most significant factor even after adjusting for age and BMI. A *T*-test between age and AUC revealed that age was directly proportional to AUC (*p* < 0.001), even when corrected for dosage (indicated as AUCmk). The AUCmk increased by 1 mg/L/h/kg every 10 years, with a coefficient of 0.1.

An important evaluation was conducted regarding BMI as a predictor of AUC, with a coefficient of 5.3, a *T*-test of 0.019, and a significant effect of BMI regardless of age. Additionally, the dose/BSA ratio was a relevant factor for AUC, with a coefficient of 0.2 and a *T*-test of 0.001. Clinicians considered this a highly predictive and user-friendly parameter that warrants further investigation. Analysis of correlation indices between concentrations at different sampling times and AUC showed that concentrations at 2, 4 and 6 h were the most predictive. The single most predictive concentration was at T2, which had a *T*-test of <0.0001 but showed the lowest R^2^ (0.86); this aligns with the findings of Nazon et al. [16] and closely resembles those of Estepp et al. and Wiczling et al. [21,22].

This study had limitations, including the assumption of optimal treatment compliance across the entire population. Patients undergoing transfusion and chelation therapy were not analyzed separately. Additionally, Pk parameters were measured following the administration of each patient’s individualized average daily dose rather than a standardized dose. But, while this approach reduces standardization, it better reflects real-life conditions. Moreover, a clinical study on pharmacogenetics and the correlations between variables in patients is currently underway.

## 4. Materials and Methods

### 4.1. Population Enrolled

We performed a retrospective study in 80 SCD patients treated at the Hemoglobinopathies University Center of San Luigi Gonzaga Hospital in Orbassano (Turin, Italy) enrolled between July 2023 and March 2024. The study protocol was approved by the local Ethics Committee (“Exploration study of pharmacokinetics and pharmacogenetics of the drug Hydroxyurea in patients with beta-hemoglobinopathy: non-transfusion dependent beta thalassemia and sickle cell anemia”, PKPG-Betahb). Orally expressed informed consent for the study was obtained from each enrolled subject. The study was carried out according to the routine schedule and patients’ availability during their regular follow up visit and involved patients of all ages and backgrounds to ensure a diverse population. The study included naïve or previously treated patients with HU (tablets of Siklos^®^) after a 48 h washout period to ensure that the administered drug had been completely eliminated. For all the patients, the following data were available and collected: sex, age, administered HU dose, date of initiation of therapy, BMI, BSA and ethnicity.

### 4.2. Methodology for HU Plasma Level Determination (Sampling Time Points, Extraction Technique and HPLC-UV Conditions)

Plasma HU concentrations were determined from samples before drug intake and 2, 4, 6 and 24 h after drug administration. Patient samples (collected in a lithium–heparin tube, 5 mL) were centrifuged at 50 rounds/s (Hz) for 10 min at 4 °C within 30 min from blood sampling, and plasma was stored in cryovials at −20 °C before the analysis.

HU concentrations were determined using an HPLC-UV method after derivatization with Xan as previously described [20], with partial modifications.

A stock standard solution of HU was prepared at 13.1 mM in water, aliquoted and stored at −80 °C. A stock solution of Me-U was used as an internal standard (IS) and was prepared at 232.5 mM in water and stored at −80 °C. Xan solution was prepared at 0.02 M in isopropanol.

For the calibration curve, a series of different standards at known and increasing concentrations of the analyte were prepared by appropriate dilutions of the stock standard solution in drug-free plasma in order to obtain the following concentrations: HU 0.625 μg/mL, 1.25 μg/mL, 2.5 μg/mL, 5 μg/mL, 10 μg/mL, 20 μg/mL, and 40 μg/mL. Calibration curves were performed at each analysis set to ensure quality measure. Each set of calibration curves also included a blank plasma sample and a blank plasma sample plus IS. On the day of analysis, calibration standards were freshly prepared from the HU stock standard solution (1 mg/mL in water), while patient samples were thawed. Calibration plasma samples and patients’ plasma samples were extracted at the same time, as follows:First, 200 μL of plasma samples was spiked with 50 μL of Me-U as IS;Proteins were precipitated by adding 600 μL of methanol;The derivatization of HU and Me-U was carried out by adding 100 μL of Xan and 50 μL of 1.5 M HCl to form xanthyl derivatives of HU and Me-U;The mixture was vortexed, left at room temperature while protected from light for 5 min and centrifuged at 10,000 × *g* for 10 min.

HU and Me-U derivatives were then separated using HPLC and quantified by UV detection at 233 nm. Separation was performed on a C18 column (4.6 × 150 mm, 5 μm) using a mobile phase consisting of the following:Briefly, 70% ammonium acetate 20 mM pH 6.5 and 30% acetonitrile from 0 to 10.5 min (phase A);Then 40% ammonium acetate 20 mM pH 6.5 and 60% acetonitrile from 10.51 to 16.5 min (phase B);Phase A from 16.51 to 18 min.The total analysis time was within 18 min at a flow rate of 1.000 mL/min.

### 4.3. Statistical Analysis

We present descriptive statistics of continuous variables as median and IR, mean, SD, SE, CI and min–max range. The Shapiro–Wilk test was used to assess normal distribution. Categorical variables were expressed as frequencies and percentages. Differences between variables or groups were analyzed using Student’s *T*-test. Spearman correlation was used to analyze the relationship between variables’ correlation, univariate and multivariate linear and logistic regression. A *p*-value < 0.05 is considered to be statistically significant. Statistical analyses were performed using STATA-SE software (STATA-SE v 18).

## 5. Conclusions

Despite the debilitating nature of SCD, current treatment options remain very limited. It is crucial to invest in the development of new therapies as well as in the optimization of currently available, safe and affordable treatments, such as HU drug use. The study carried out here initiated the exploration of PK-guided HU dose escalation, already investigated as demonstrated by the available scientific literature [9,10,11,12,13,15,16,17,18,20,22] but not yet well established as a useful clinical tool for patient management. We showed the importance of addressing Pk variability through TDM, a practice repeatedly shown to be necessary in this clinical field. Our goal is to make this approach more feasible and to integrate it into standard care routines. Pk can play a key role in enhancing the standard of care provided to ameliorate patients’ quality of life and their management.

## Figures and Tables

**Figure 1 pharmaceuticals-17-01386-f001:**
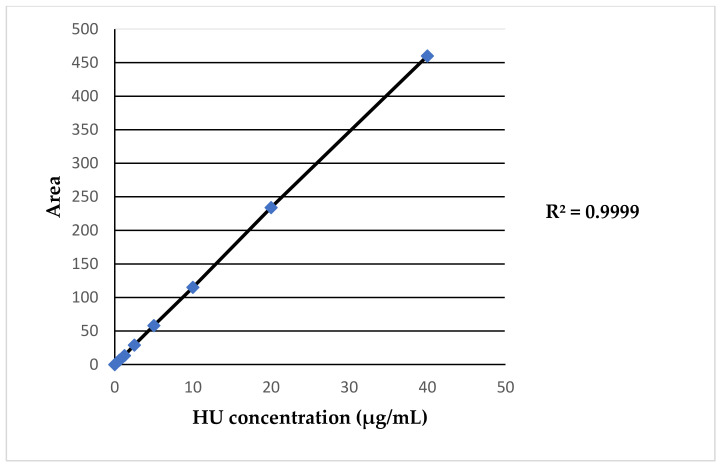
Example of HU calibration curve.

**Figure 2 pharmaceuticals-17-01386-f002:**
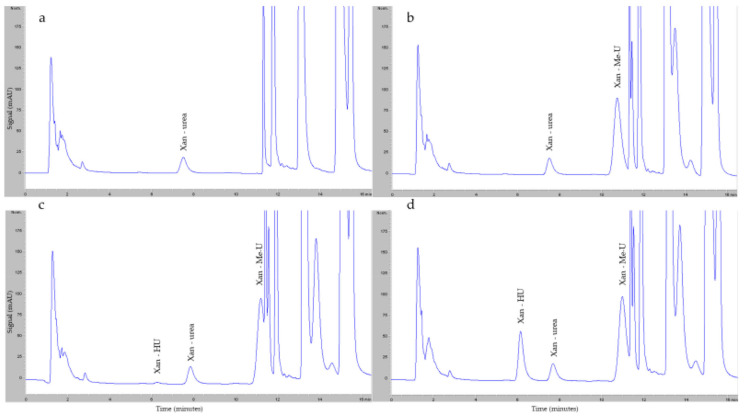
Example of HU peaks. Representative chromatograms of (**a**) blank plasma sample supplemented with Xan; (**b**) blank plasma sample supplemented with Xan and Me-U; (**c**) the lowest point of the calibration curve: HU 0.625 μg/mL; (**d**) the highest point of the calibration curve: HU 40 μg/mL.

**Figure 3 pharmaceuticals-17-01386-f003:**
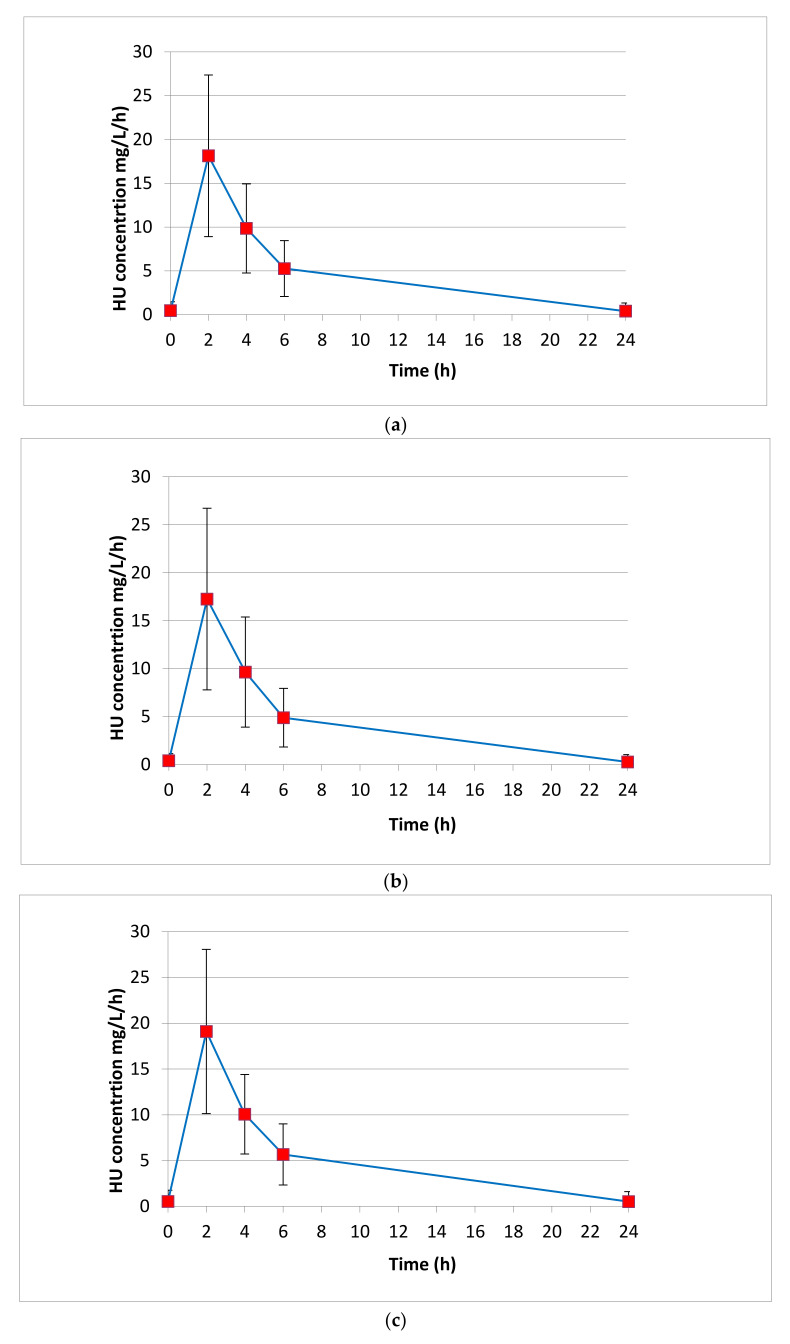
HU AUCs expressed by the median values obtained for each sampling point (0 prior to drug administration; 2, 4, 6, 24 h after drug intake). (**a**) Median AUC calculated for all patients (N = 80). (**b**) Median AUC calculated for female patients (N = 42). (**c**) Median AUC calculated for male patients (N = 38).

**Table 1 pharmaceuticals-17-01386-t001:** Sex-disaggregated age (in years) of patients.

	N (%)	Median	IQR
Female	42 (52.2%)	11.6	7.2–55.1
Male	38 (47.8%)	17.4	9.4–73
Total	80	13.4	7.8–73

**Table 2 pharmaceuticals-17-01386-t002:** Population ethnicity.

Ethnicity	Patients (N)	Patients (%)
African	58	72.5%
Caucasian	17	21.3%
Other	5	6.3%

**Table 3 pharmaceuticals-17-01386-t003:** Pk parameters.

Variable	Patients (N)	Median	IQR
AUC (mg/L/h)	80	101.3	72.5–355.9
T_1/2_ (h)	69	2.7	2–62.2
K_e_	69	0.3	0.1–0.8
C_Max_ (mg/L/h)	80	17.7	12–53.1
T_Max_ (h)	80	2	2–4

**Table 4 pharmaceuticals-17-01386-t004:** HU AUCs (mg/L/h).

	N (%)	Median	IQR
Female	42 (52.2%)	97.4	71.9–336.8
Male	38 (47.8%)	117.9	73.6–355.9
Total	80 (100%)	101.3	72.5–355.9

**Table 5 pharmaceuticals-17-01386-t005:** AUC regressed for BMI and dose/BMI.

		AUC (mg/L/h)		
	Coefficient	SE	CI	*T*-test
BMI (kg/m^2^)	5.3	2.2	0.1; 9.8	0.019
Dose/BMI	1.4	0.6	0.3; 2.5	0.015

**Table 6 pharmaceuticals-17-01386-t006:** AUC regressed for both age and BMI.

		AUC (mg/L/h)		
	Coefficient	SE	CI	*T*-test
BMI (kg/m^2^)	4.3	2.0	0.3; 8.3	0.035
Age (years)	0.6	0.7	−0.7; 1.9	0.378

**Table 7 pharmaceuticals-17-01386-t007:** Average dose/BSA.

		Dose/BSA (mg/m^2^)		
Patients (N)	Mean	SD	Min	Max
80	604.5	135.7	299.3	919.7

**Table 8 pharmaceuticals-17-01386-t008:** AUC and dose/BSA.

		AUC (mg/L/h)		
	Coefficient	SE	CI	*T*-Test
Dose/BSA	0.2	0.1	0.07; 0.29	0.001

**Table 9 pharmaceuticals-17-01386-t009:** AUC of steady-state patients regressed for ethnicity.

		AUC (mg/L/h)		
	Coefficient	SE	CI	*T*-Test
African	−61.7	19.4	−100.5; −23.0	0.002
Other	−72.2	38.0	−148.0; 3.6	0.062

**Table 10 pharmaceuticals-17-01386-t010:** AUC corrected for dosage of steady-state patients and regressed for origin.

		AUCmk (mg/L/h)		
	Coefficient	SE	CI	*T*-Test
African	−5.3	1.0	−7.3; −3.3	<0.001
Other	−5.2	2.0	−9.1; −1.2	0.011

**Table 11 pharmaceuticals-17-01386-t011:** AUC results comparison for patients at MTD.

	N	Mean	Range	SD	CI	*T*-Test
			Min–Max			
AUC 1 (mg/L/h)	63	115.7	61.5–317.8	34.0	107.1; 124.3	
AUC 2 (mg/L/h)	65	118.8	42.4–355.9	66.6	102.4; 135.4	
Δ		−3.2			−21.8; 15.4	0.733

(AUC 1 = mean value obtained by Dong et al. [18]; AUC 2 = mean value obtained by adapted methodology).

## Data Availability

The original contributions presented in the study are included in the article; further inquiries can be directed to the corresponding author/s.

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
