# Peer review of "Hydroxyurea Pharmacokinetic Evaluation in Patients with Sickle Cell Disease"

_pharmaceuticals, 2024, doi:10.3390/ph17101386_

Round 1

Reviewer 1 Report

Comments and Suggestions for Authors

The main purpose of the present study was to develop an HPLC-UV method to measure HU plasma concentration. It was also determined that the developed method was to investigate the main Pk parameters that occur after the administration of a single dose of HU in SCD patients and to evaluate possible differences in HU AUC plasma levels between male and female patient populations. The study is of great importance for the field of clinical pharmacology. However, in the literature searches, many more detailed studies on HU pharmacokinetics and pharmacodynamics were found. (For example: Paule, I., Sassi, H., Habibi, A. et al. Population pharmacokinetics and pharmacodynamics of hydroxyurea in sickle cell anemia patients, a basis for optimizing the dosing regimen. Orphanet J Rare Dis 6, 30 (2011).) When the article was examined in detail, no detailed data presentation was found regarding the development of a new method. In addition, it was determined that the results obtained in the presented article were discussed as needing more studies or studies are ongoing, and therefore their importance was not discussed sufficiently. However, I believe that comparing the differences in the results with the studies previously conducted with HU and writing them down by focusing on a possibility would reveal the importance of the study more. For these reasons, I believe that the subject of the presented study has been studied many times before and that the importance of the results obtained from the article is prevented from being sufficiently revealed because it was not written appropriately.

Reviewer 2 Report

Comments and Suggestions for Authors

Review pharmaceuticals-3197758. Hydroxyurea pharmacokinetic evaluation in Sickle Cell Disease patients

It is an interesting article. The authors try to correlate different variables with hydroxyurea exposure in patients with sickle cell disease.

It is a very transferable objective, applied directly to the treatment of patients.

They develop a method to analyze plasma samples in HPLC with a good r2.

They explore the current data to find covariates that can explain the variability of pharmacokinetics and find a correlation between AUC and BMI.

I consider the study of pharmacokinetics of this type of treatments a vital practice.

I recommend accept this paper to publication.

Key words

If a word is in the title you cannot include as keyword, please delete and replace those words in the key words section.

Results

Line 246-247: “No patient started treatment before he/she was 2 years old, while the average age at the beginning of treatment was 10.9 ± 10.4 years (min = 1.3; max = 49.3).”

Figures:

The Y axis should start at 0

Something is missing in this sentence. Patient with 1.3 years is younger than 2 years and a mean of 10.9 with an SD of 10.4 seems incorrect also. (Check possible 0s)

Reviewer 3 Report

Comments and Suggestions for Authors

Dear authors 

thanks for your efforts 

Hydroxyurea pharmacokinetic evaluation in Sickle Cells

Lines

Present

Comments

Abstract

Our

Better to use clear words

Abstract

Direct correlation

Do you mean positive

Abstract

our paper

???

64

prevalence

Incidence ???

135

sex and gender

The author didn’t use gender in the table

Table 1

Mean 14.1 while SD 11

This mean the data not normaly disterbured – medain should used

Table 1

Mean 21 while SD 15.8

This mean the data not normaly disterbured – medain should used

244

Patients in steady state

treatment have been taking HU for 7.3 ± 5.8 years

SD in large number

Table 2

Mean 122.1 while SD 70

This mean the data not normaly disterbured – medain should used

Table 8

P value 0.0014

0.001

442

4. Materials and Methods

Should be after introduction

504

Student’s T-test

Should used in normaly disterbuted data- but data in the article is not normal y disterbuted

552

[6] Ballas S. K. (2020)

Many references not equaly formatted

Round 2

Reviewer 1 Report

Comments and Suggestions for Authors

It was determined that the requested corrections were made by the author. I believe that the article is suitable for publication in its current state. It was determined that the revised article was restructured in an understandable and clear manner. It was determined that the hypothesis was defined in an understandable manner after the requested corrections. It was seen that the scientific design of the article was expressed in a more scientifically sound manner and was reorganized with renewed references. It was determined that minor corrections were made in the material and method section of the article and that the results obtained from the study were rewritten in an appropriate and consistent manner to reveal the importance of the article. In addition, it was determined that the discussion and conclusion sections of the article were rewritten in line with the purpose of the article and the results obtained and were interpreted appropriately.

Author Response

Dear reviewer,

thank you for your appreciation.

Reviewer 3 Report

Comments and Suggestions for Authors

Dear author 

thank you for your corrections 

Author Response

(The authors gave the same response as above.)
